

# Surface mass balance modelling of the Juneau Icefield highlights the potential for rapid ice loss by the mid-21st century

Ryan N. Ing[1], Jeremy C. Ely[2], Julie M. Jones[2], Bethan J. Davies[3]

[1]School of Geosciences, University of Edinburgh, Edinburgh, EH8 9XP, UK
[2]Department of Geography, University of Sheffield, Sheffield, S10 2TN, UK
[3]School of Geography, Politics and Sociology, Newcastle University, Newcastle upon Tyne, NE1 7RU, UK

*Correspondence to*: Jeremy C. Ely (j.ely@sheffield.ac.uk)

**Abstract.** Plateau icefields are large stores of freshwater, preconditioned to enhanced mass loss due to their gently sloping accumulation areas. Accurately modelling the mass-balance of these icefields is therefore vital for obtaining projections of their future contribution to sea-level rise. Here, we use the COupled Snowpack and Ice surface energy and mass-balance model in PYthon (COSIPY) to simulate the historical and potential future mass balance of the Juneau Icefield, Alaska – a high elevation (>1200 m) plateau icefield. We force the model with dynamically downscaled climate simulations, pertaining to both the past and potential future (RCP 8.5) conditions. The rich dataset of surface mass balance observations of the Juneau Icefield allows us to tune COSIPY, providing confidence in our future predictions and highlighting changes to the icefield between the years 1980 and 2019. Icefield-wide negative mass balances were simulated from the start of the 21st century, as many glaciers transitioned from positive to negative mass-balances. Under RCP8.5, the model simulates increasing negative mass balance across Juneau Icefield, with the entire icefield potentially displaying a negative mass balance by the mid-21st century. This simulated loss of accumulation is driven by increased temperatures and reduced amounts of snowfall, exposing more of the icefield to thinning. Ice thinning is likely to be exacerbated by the exposure of ice to melting across the plateau surface, and prolonged melt may lead to an increase in disconnections, splitting glaciers between their accumulation and ablation areas at icefalls. The similar hypsometry of other high latitude plateau icefields and ice caps may mean that similar processes will act to determine their potential fate in our changing climate.

## 1 Introduction

In our warming world, the stores of frozen freshwater contained within the ice of glaciers and ice sheets are depleting, contributing to global sea level rise (Zemp et al., 2019; Edwards et al., 2021). Although glaciers contain but a small proportion of global ice volume (~1%), they are disproportionally contributing to sea level rise when compared to the larger ice sheets (Marzeion et al., 2020; Hugonnet et al., 2021). Glaciers in Alaska are the largest source of released freshwater, having produced one-third of the sea-level contribution from glaciers between 1961 and 2016 (Zemp et al., 2019). Global-scale projections of future change indicate that glaciers will continue to disproportionally contribute to sea level rise, and that glaciers in Alaska will remain the largest suppliers of meltwater (Hock et al., 2019; Edwards et al., 2021). In Alaska, the



majority of glacial ice is held within its several icefields and ice caps situated along the mountainous coastal regions (Davies et al., 2022). Though global-scale projections show a linear loss of ice mass with temperature increases (Rounce et al., 2023), the hypsometry of plateau icefields, such as Juneau Icefield (Fig. 1), makes them susceptible to rapid ice loss through three processes: i) once the equilibrium line altitude (ELA) rises to the elevation of the plateau, small increases in ELA lead

to a rapid decrease in accumulation area and widespread thinning (Böðvarssson, 1955; Bolibar et al., 2022); ii) ELA rises cause the exposure of bare ice, which has a lower albedo than snow, promoting melting through an albedo-feedback (Johnson and Rupper, 2020); iii) dynamic thinning across icefalls flowing over steep topography can lead to the disconnection of glacier tongues from the plateau, leaving them devoid of an accumulation area, with glacier tongues left only to ablate (Davies et al., 2022). Thus, reliable projections of the future mass balance of Alaskan icefields and ice caps are

critical for projecting future sea-level rise.

Among the largest of the Alaskan icefields is Juneau Icefield (Fig. 1). Situated within the Coast Mountain Range, and on the border between Alaska and British Columbia, Juneau Icefield consists of a high elevation (>1200 m) plateau which is drained by numerous outlet glaciers (Sprenke et al., 1999; Davies et al., 2022). Numerous observational assessments of the mass-balance of the Juneau Icefield have been conducted. The most comprehensive of these comes from the Juneau Icefield

Research Program (JIRP), which has collected direct stake-based field measurements of accumulation and ablation at Lemon Creek and Taku glaciers annually since 1947 (LaChapelle, 1954; Wilson, 1959; Miller, 1975; O'Neel et al., 2019; McNeil et al., 2020). Geodetic methods have also provided estimates of the surface mass balance (SMB) of Juneau (Larsen et al., 2007; 2015; Larsen, 2010; Berthier et al. 2010; 2018; Melkonian et al., 2014). Of these, Larsen et al. (2015) found that all glaciers had a negative mass balance, with the notable exception of Taku glacier. This is consistent with field-based measurements,

but subsequent measurements have highlighted that Taku has also switched to a negative mass balance, initiating recession (McNeil et al., 2020).

Although observations of Juneau Icefield are rich, modelling icefield-wide mass-balance remains challenging. The mountainous terrain and maritime climate of Juneau makes modelling the climate in the region difficult as it requires high-resolution climate models to capture orographic processes (Bozkurt et al., 2019). Global-scale projections of mass-balance,

of course, include Juneau within their domain (e.g. Hock et al., 2019; Rounce et al., 2023). Whilst global assessments are of upmost importance, the scale of these experiments means that they necessarily require simplified models of glacier mass-balance, to drive ice flow models. Examples of mass balance models used include positive-degree day models (e.g. Marzeion et al., 2012; Huss and Hock, 2015) and simplified energy balance models (Giesen and Oerlemans, 2013). Such models neglect the full energy balance of glacier surfaces. Furthermore, coarse resolution global climate models are used as input

into these models. Although often statistically downscaled to higher resolutions, such approaches underrepresent orographic processes. For Juneau, this means that detailed spatial patterns of mass-change are unobtainable from global-scale models. The first, and thus far only, previous study to specifically model the mass-balance of the Juneau Icefield is that of Ziemen et al. (2016). In Ziemen et al. (2016), a dynamically downscaled climate model (CCSM dynamically downscaled to 20 km resolution) was used as input to a positive degree-day model. Ziemen et al. (2016) found that a fit to mass-balance and ELA



observations was only obtainable by manually introducing a precipitation gradient across the icefield; a choice designed to compensate for the lack of orographic processes producing a rain shadow effect in the model, but one that lacks physical realism (Roth et al., 2018). Ziemen et al. (2016) also found that statistically downscaled climate data also provided a poor fit to observations, due to simplistic elevation dependent downscaling methods for precipitation, despite being a higher resolution (2 km).

In this study, we aim to improve the SMB modelling of the Juneau Icefield. To achieve this goal, we will utilise the high-resolution climate model simulations of Southeast Alaska of Lader et al. (2020), and a more complex mass-balance model than has previously been applied to Juneau – the COupled Snowpack and Ice surface energy and mass-balance model in PYthon (COSIPY; Sauter et al., 2020). The long observational dataset from the Juneau Icefield (e.g. O'Neel et al., 2019; McNeil et al., 2020) provides an ideal opportunity to calibrate the model, and assess the accuracy of historical simulations.

We then perform projections of the near future (2031 to 2060) mass balance of the Juneau Icefield. We use these experiments to examine potential drivers of future change, examine potential changes to surface meltwater and runoff, and to shed-light on its potential fate.

## 2 Methodology

Our overall approach is summarised in Fig. 2, with subsequent sections providing more detail on each stage. Datasets

pertaining to the topography, glaciology and climate (Lader et al., 2020) of the Juneau Icefield were input into COSIPY (Section 2.1). An important distinction is between the reanalysis climate model (Climate Forecast System Reanalysis, CFSR, Saha et al., 2010), which utilises observations to generate a climate close to reality, and "free-running" climate models (Geophysical Fluid Dynamics Laboratory Climate Model version 3, GFDL-CM3, Donner et al., 2011, and Community Climate System Model version 4, NCAR-CCSM4, Gent et al., 2011). Simulations forced by the CFSR allow evaluation of

simulations with COSIPY without the complexity of biases inherited from the model forcing. Free running simulations are forced by time series of atmospheric greenhouse gas concentrations, solar variability and volcanic activity, but are unconstrained by observations.  These simulations should capture the part of climate variability that is due to these external forcings, hence should produce general climatic patterns and trends, but not necessarily the exact timing of individual events. All data was projected onto a single grid system, and the climate datasets were bias-corrected (Section 2.2). The reanalysis

climate data was used to drive a perturbed parameter ensemble of simulations. Comparison of these reanalysis driven simulations to mass-balance observations was used as the basis of model optimisation (Section 2.3). The best-fitting parameter combination was used to drive mass-balance simulations from free-running historical climate simulations, to evaluate whether these models adequately conform to the broad climatic patterns and trends across the Juneau Icefield, before they were utilised in projections. As such, we produced three sets of simulations of the reference SMB of Juneau; an

evaluation simulation spanning 1981 to 2019 which utilised climate reanalysis data, a historical simulation utilising free running climate model simulations, and projections of future SMB under RCP 8.5 between 2031 and 2060.



## 2.1 COSIPY

COSIPY, an updated version of COSIMA (COupled Snowpack and Ice surface energy and mass-balance model in MAtlab), is an open-source mass balance model that provides a flexible framework for modelling snow and glacier mass changes (Huintjes et al., 2015; Sauter et al., 2020). Since its release it has been used for various applications; from simulating the behaviour of Schiaparelli Glacier, Chile, during the Little Ice Age (Weidemann et al., 2020) to investigating the sensitivity of Halji glacier in the Himalayas (Arndt et al. 2021). Its modular structure lends itself to being edited for use on a wide array of applications and locations. Additionally, it has proven to be effective when forced with climate data that has been dynamically downscaled using WRF (Blau et al., 2021), which we will be utilizing in this study. Since COSIPY is a one-dimensional model, it neglects any lateral and basal processes such as snowdrift or lateral mass and energy fluxes. As such, COSIPY represents a middle ground in complexity when compared to other SMB models. Here, we include only a brief description of the model. For a full description of COSIPY, see Sauter et al. (2020).

COSIPY consists of two coupled models: a surface energy-balance model and a multi-layer subsurface snow and ice model. The model assumes full conservation of mass and energy in the snowpack, with the total mass balance consisting of the SMB and internal mass balance (IMB). The SMB is calculated through the sum of surface accumulation (snowfall and deposition) and ablation (surface melt and sublimation) processes. The calculated surface melt is used as an input to the subsurface model which accounts for many sub-surface processes including meltwater percolation, retention, refreezing and sub-surface melt – all of which are resolved in a vertical layer structure.

Snowfall at each grid point is calculated using a logistic transfer function combining precipitation, temperature and relative humidity (Hantel et al., 2000). Fresh snow is added in the model only if snowfall exceeds 0.001 m per day. Snow albedo ($a_{snow}$) is parameterised using the approach by Oerlemans and Knap (1998), whereby

$$a_{snow} = a_{firn} + \left(a_s - a_{firn}\right) \exp\left(\frac{s}{\tau*}\right), \tag{1}$$

where $a_{firn}$ and $a_s$ are the albedo values for firn and fresh snow as set by the user. $\tau*$ is the albedo timescale and defines the time taken for the snow albedo to drop from fresh snow to firn, $s$ defines the number of days since the last snowfall. The model also considers the effect of snowpack thickness on albedo. After falling, snow settles and compacts during metamorphism, causing the density of the underlying snowpack to increase, impacting thermal conductivity and liquid water content. The rate of density change of each snow layer is calculated using the method described in Boone (2004), where the densification happens through the overburden of pressure (Essery et al., 2013). At the surface, turbulent fluxes are parameterized based on a flux-gradient similarity theory. A linear roughness length with time is used to calculate the turbulent fluxes using the method from Mölg et al. (2012).

COSIPY was initialised with the default parameterisations and values of Blau et al. (2021), and later optimised via the procedure described in Section 2.3. A spin-up period of an entire month of August, the month with the lowest average snowfall, was conducted for all model runs to reach a state of equilibrium before calculating the SMB.



## 2.2 Input data to COSIPY

Two types of input data are required by COSIPY: static data and climate data. Static data does not change during a model simulation. This consists of a shapefile of the region of interest and a digital elevation model (DEM). For our region of interest, we use the glacier outlines of Juneau Icefield in 2019 from Davies et al. (2022). The DEM used in this study is derived from the Shuttle Radar Topography Mission, which has a global coverage of 1-arc second (~30 m). This was aggregated to the coarser resolution of 600 m, which formed the model grid. From the aggregated DEM, other input grids of

surface slope, aspect, hillshading and an ice mask were calculated using the Geo-spatial Data Abstraction Library (GDAL). COSIPY does not account for the evolution of an ice surface or margin during a model simulation, hence, the SMB presented here should be considered as reference SMB, according to the definitions of Cogley et al. (2010).

To force COSIPY, we utilise the dynamically downscaled climate models of Lader et al. (2020). In Lader et al. (2020), a selection of climate models were dynamically downscaled to an hourly temporal and 4 km spatial resolution using the

Weather and Research Forecasting model version 4 (WRF) (Skamarock et al., 2019). At this resolution, the model is able to accurately resolve the effect of mountainous topography and reduce the amount of parameterisations needed to model processes such as cumulus convection.

The original data used by Lader et al. (2020) for dynamical downscaling consists of output from three models. One reanalysis model (CFSR; Saha et al., 2010), is used here for evaluation simulations of past SMB between 1980 and 2019.

This was chosen by Lader et al. (2020) as a target for dynamical downscaling as it is one of the top performing reanalysis models for southeast Alaska (Lader et al., 2016). Furthermore, output from the dynamically downscaled CFSR model shows good agreement with observations from the automatic weather station at Juneau airport, albeit with a slight positive precipitation bias likely due to the large range of topography within the grid cell of Juneau airport (Lader et al., 2020). Two dynamically downscaled climate models (GFDL-CM3, Donner et al., 2011, and NCAR-CCSM4, Gent et al., 2011), provide

both historical runs for 1980-2010, and projections for the period 2030 to 2060. These two models were chosen by Lader et al. (2020) as they routinely rank in the top five of all CMIP5 models for Alaska (Walsh et al., 2018). Usefully, the two climate models also have different climate sensitivities – NCAR-CCSM4 has a low climate sensitivity, GFDL-CM3 has a high climate sensitivity (Flato et al., 2013). Henceforth we refer to NCAR-CCSM4 as CCSM and GFDL-CM3 as GFDL. The use of these specific model datasets is outlined in Fig. 2.

The data provided by Lader et al. (2020) provides dynamically downscaled projections of climate under the RCP8.5 emissions scenario. Numerous recent studies have emphasised caution on the overuse of the RCP8.5 scenario, with recent work suggesting it is unlikely (Hausfather and Peters, 2020; Burgess et al., 2020). However, as highlighted by Lader et al. (2020), since our projections are focussed only on the short-term future (to 2060), the choice of climate scenario is largely irrelevant. This is due to the atmospheric residence time of the greenhouse gases and aerosols already emitted, with many

models in CMIP5 predicting similar increases in global air temperatures for all RCP emissions until the year 2060 (Overland et al., 2014).



Despite the dynamically downscaled products from Lader et al. (2020) having a high spatial resolution of 4 km, many glaciers of Juneau Icefield have a width of less than 1 km, creating the need for further downscaling or interpolation. For 2 m air temperature ($T_2$) and surface pressure ($p$), the climate data grid point of each variable closest to the COSIPY grid point was first selected before applying the chosen downscaling method. $T_2$ was downscaled using a constant lapse rate of 5 K km$^{-1}$, chosen through analysis of several AWS sites in McNeil et al. (2020). Surface pressure was downscaled using the barometric formula. The precipitation, incoming longwave and shortwave radiation were all bi-linearly interpolated onto the COSIPY model grid. Wind speed was interpolated onto the model grid in the same manner, after being derived from wind speed components at 10 m, assuming a logarithmic wind profile (Arndt et al. 2021). The incoming shortwave radiation was additionally adjusted after being interpolated to the model grid by a radiation model based on Wohlfahrt et al. (2016). The model corrects the incoming shortwave radiation by taking into account the effect of shadowing from nearby terrain at each grid-point throughout the year.

The final step in preparing the climate input was to conduct bias correction of the downscaled global climate model data (CCSM and GFDL). Since these models are only forced by radiative and aerosol forcings, they do not replicate past inter-annual variability well, but should capture longer term trends. Here, we used empirical quantile mapping based on the method used in Amengual et al. (2012). Empirical quantile mapping adjusts the cumulative distribution function of the future projected climate data by adding the mean regime shift and deviation of the climate models to the quantiles of the observed dataset. Values of certain variables that are outside of the range of the observational dataset are corrected using constant extrapolation. Additionally negative precipitation and radiation values are set to zero. Relative humidity is also bounded from 0 to 100. For simplicity, the wet day frequency and drizzle threshold of the bias corrected climate data was not adjusted.

### 2.3. Model optimisation

For COSIPY to accurately simulate the SMB of Juneau Icefield, there are several model parameters that need to be optimised. These are summarised in Table 1. To optimise the model, we use a modelled glacier-wide mass balance dataset provided by the Juneau Icefield Research Project (JIRP). This model combines SMB data collected between July and August from numerous snow pit and ablation stake locations on Taku and Lemon Creek glaciers, with glacier hypsometry data to overcome the lack of measurements in ablation areas, missing data and inconsistencies in the timing of data collection. The associated uncertainty with this mass balance model is ± 0.45 m w.e. a$^{-1}$.

Initial values for model optimisation were selected from the literature and previous uses of COSIPY. The albedo ranges were specifically selected to not overlap with each other. A Latin Hypercube was used to generate a random sample of 100 values for each of the 10 parameters, with the sampling evenly distributed across the entire parameter space. COSIPY was then run in an ensemble of 100 model runs, forced with CFSR, each with a different set of parameters. The model simulations were then scored against the observations of SMB from JIRP using the Root Mean Square Error (RMSE) and the coefficient of determination ($R^2$).



A series of 100, 6-year long (2003 to 2008), runs for Taku Glacier were conducted as the optimisation set. The optimisation process was not conducted on Lemon Creek Glacier due to its smaller size and location on the moister south-west side of the icefield, which initial testing showed acted to decrease the overall SMB of the model and the precipitation used. Fig. S1 shows the results from the optimisation process. The parameter values of the top 10 runs were investigated to ensure that the process was finding sensible parameter combinations. The parameter with the largest effect on simulations was the multiplication factor for total precipitation, which was between 0.7 and 0.8 for all of the top 10 performing model runs.

The best scoring parameter combination is shown in Table 1. These parameters were used to conduct a longer model simulation for both Taku and Lemon Creek glaciers (1981-2019), to investigate whether the model would produce realistic results outside of the optimisation period. Results from this simulation are shown in Fig. 3. The model matches the JIRP observations for Taku glacier in and outside the optimization period, with the COSIPY simulations matching the observed inter-annual variation well (Fig. 3a). The model appears to fit less well for the Lemon Creek glacier. This is likely owing to the glaciers' relatively small area, which is less than three grid cells of the dynamically downscaled WRF data. Over the whole icefield domain, this is likely to have less of an effect. A slight positive bias was noted for the simulations of both glaciers. This might be due to the model using a shapefile of the glaciers area in 2019 and therefore potentially missing grid points in the ablation areas. Nevertheless, with these results we can be confident that the COSIPY model has been optimized and is performing well for the Juneau Icefield region.

## 3 Results

### 3.1 Evaluation simulations of surface mass balance from reanalysis (1981-2019)

Across Juneau icefield, the reanalysis simulation from CFSR-COSIPY displayed a non-statistically significant warming trend of +0.02 °C decade$^{-1}$, with a mean annual temperature across the domain of -3.27 °C. Unsurprisingly, elevation is the strongest control upon temperature. A lapse rate of 6 K km$^{-1}$ positions the lowest mean annual air temperatures in the centre of the plateau (~ -8 °C) and the highest at low elevation tongues of outlet glaciers (Fig. 4a). According to the CFSR simulations, the mean annual total precipitation across the icefield is 3060 mm a$^{-1}$, with a maximum occurring in the southwest portion of the icefield and a minimum occurring at the tongue of Llewellyn Glacier (Fig. 4b). Snowfall follows a similar spatial pattern, the difference being a more concentrated maximum in the southwest portion of the icefield, and little to no snowfall at the tongues of outlet glaciers due to higher temperatures (Fig. 4c). Both precipitation and snowfall have a non-statistically significant negative trend during the historic period (Table S2). The mean annual incoming shortwave radiation across the icefield in the CFSR simulation is 110 W m$^{-2}$. Likely due to the southwest of the icefield being more moist and cloudier, results from the CFSR modelling show a strong gradient across the icefield of incoming shortwave radiation with lower values in the southwest (~ 95 W m$^{-2}$) and higher values in the northeast (~130 W m$^{-2}$) (Fig. 4d).

The SMB of the whole icefield for the historic period is summarised in Fig. 5a. The reanalysis-driven simulation (CFSR-COSIPY) showed an average annual SMB of -0.08 m w.e. a$^{-1}$, with a non-statistically significant trend of -0.02 m w.e. a$^{-2}$.



Within the accumulation area of the plateau, mass balance is often greater than 2 m w.e. a$^{-1}$, whilst at the low elevation ablation zone of the glaciers the mass balance is below -4 m w.e. a$^{-1}$. The icefield wide estimates of mass balance produced by CSFR-COSIPY are within the range of estimates of specific mass balance from previous studies, and very close to those

of Ziemen et al. (2016) (Table 2). Furthermore, if we subset our modelled mass balance to the time period to match that of the most recent observational estimates from Berthier et al. (2018), our simulated mass balance is within observational uncertainty (Table 2). The CFSR driven run also predicts a negative trend in mass balance for many of Juneau Icefield's outlet glaciers over the evaluation period (Fig. 5b). These per-glacier simulations of mass balance are in good agreement with estimates for nine glaciers derived from laser altimetry (Larsen et al., 2015), although there is a slight positive bias

(Table S1). The model simulates that many of these nine glaciers had a slight positive SMB at the start of the evaluation period (Fig. 5b); an exception to this is Gilkey glacier, which had a negative SMB throughout the majority of the simulations (Fig. 5b). In the simulations, the SMB of the icefield begins to be negative around the late 1980s, as does the SMB of the Mendenhall, Gilkey and Lemon Creek glaciers. Consistent with observations (McNeil et al., 2020), the SMB of Taku glacier begins to decrease in the 2010s (Fig. 5b). The model also indicates that Willison and Llewellyn glaciers exhibit a negative

SMB from ~2010 onward (Fig. 5b).

During the evaluation simulation period, the icefield-wide ELA was ~1360 m on average, with individual glacier ELAs ranging between 1150 m to 1640 m. The average accumulation area ratio (AAR) for this period is 56%, and covers the majority of the plateau. The spatial pattern of simulated ELA using the CFSR reanalysis data, compares favourably with ELA observations (Ziemen et al., 2016; Fig. S4). The lowest ELAs occurred at the southerly glaciers (e.g. Taku,

Mendenhall, Tulsequah), likely due to the higher accumulation rates at these glaciers. Across the nine major glaciers identified in Fig. 5b, a statistically significant positive trend in ELA of 12.1 m a$^{-1}$ on average occurred. At Lemon Creek glacier, the ELA was often incalculable, with all grid points having a negative SMB on multiple years from 2000 onwards.

### 3.2 Historical simulations of surface mass balance from climate models (1980-2010)

The evaluation simulation presented in Section 3.1 was driven by climate reanalysis data (CFSR) that is constrained by

observations, and thus represent our best estimate of reality. However, to obtain projections under future climate, observationally unconstrained simulations are required, forced by scenarios of future atmospheric greenhouse gas concentrations. To ascertain whether the two climate models studied here, GFDL and CCSM, were able to somewhat simulate realistic patterns of climate pertinent to mass balance processes, we first conducted simulations in COSIPY driven by the modelled historical climates from these two models covering the period 1980-2010. These historical climate

simulations were forced by past atmospheric greenhouse gas concentrations, solar variability, and volcanic forcing.

The mean values of the free-running climate models, GFDL (Fig. S2) and CCSM (Fig. S3), are statistically similar to that of the evaluation CFSR simulation shown in Fig. 4 (Table S2). However, the trends for some variables differ between model runs (Table S2). The higher sensitivity GFDL model shows a significant increase in temperature and decrease in annual precipitation and snowfall over the period 1980-2010, whilst the lower sensitivity CCSM model only shows a significant



decrease in snowfall over this period (Table S2). This led to the two models producing a spread of simulated SMB over the historical period (Fig. 6). As expected, the free running models do not match the interannual variability of the reanalysis simulations. However, when taking the mean across the whole period, each individual model is within ±0.45 m w.e. a$^{-1}$ (the uncertainty of the JIRP mass balance model; Section 2.3) of the CFSR evaluation simulation (Fig. 6), and the mean of the two free-running models and the evaluation simulation is statistically signficant (as revealed by a t-test). If the mean of the

two models is taken, the SMB is within ± 0.03 m w.e. a$^{-1}$ of the CFSR evaluation simulation (Fig. 6). Furthermore, the spatial distribution of SMB from the mean of the two model runs, matches closely with the CFSR driven simulations (Fig. S5). This gives confidence in the two models for providing projections of future SMB.

### 3.3 Future surface mass balance of the Juneau Icefield (2031-2060, RCP 8.5)

Here we discuss the SMB projections using the CCSM and GFDL models under RCP.8.5. Our justification for the use of

RCP8.5 is described in Section 2.2. The change in modelled climate between the means of the historical (1980-2010) and future (2031-2060) simulations of each model is shown in Fig. 7. By considering the future changes as differences between the historical period and future simulations for each respective model, then this accounts for any model bias (assuming the bias remains constant). Both models project an increase in temperature everywhere on the icefield, with a Mann-Kendall test of the annual averages showing a statistically significant trend for the future period of 2031-2060 at most grid points, except

for at some cells on glacier tongues. GFDL predicts a mean increase of 3.5°C, whilst CCSM predicts a mean increase of 1.01°C. There are spatial differences between the two models. The GFDL simulation shows the largest increases at the tongues of glaciers in the north and northeast, most notably at Llewellyn Glacier (Fig. 7a), whilst the CCSM simulation indicates that the tongues of glaciers in the south and west will have the largest temperature increases, most notably at Taku Glacier (Fig. 7e). Both models project a decrease in annual total precipitation almost everywhere on the icefield (Fig. 7b and

f). This is specific to the icefield domain, as precipitation across the rest of southeast Alaska is predicted have a positive trend under both models due to a deepening of the Aleutian Low (Gan et al., 2017; Lader et al., 2020). Both models also project a reduction in snowfall across Juneau Icefield (Fig. 7c and g). Little change is projected to occur in incoming shortwave radiation, though both models show a slight increase at the tongues of Llewellyn Glacier and Tulsequah Glacier. The projected future (2031-2060) SMB under the RCP8.5 emissions scenario is shown in Fig. 8. Both models display an

increasingly negative annual SMB despite their different climate sensitivities (Fig. 8). These simulations compare well with the projections assembled in Hock et al., (2019) for Alaska, who collated models that used the same RCP, but drove their mass balance models with general circulation models. This similarity also occurs despite our simulations lacking a consideration of changing glacier hypsometry due to ice flow. The mean annual SMB of the two models of -1.52 m w.e. a$^{-1}$ (2031-2060), represents a decrease of 1.41 m w.e. a$^{-1}$ when compared to the mean annual SMB of the evaluation simulations

(1980-2010). The mean trend in the annual SMB for the average of the two models over 2031-2060 was -0.064 m w.e. a$^{-2}$, which was found to be statistically significant (p=0.0001). For the mean of the two models, the accumulation area greatly reduces (Fig. 8b), especially towards to end of the simulations (not shown). When considering the mean of the two models,

there are multiple years towards the end of the simulation where no grid points on the icefield have a positive SMB (not shown). It is worth noting that is despite a positive bias of both models in past simulations, which may have transferred to

these predictions. The largest decreases in SMB are predicted to be in the accumulation areas on the icefield plateau, particularly in the south-west catchment zones of Norris, Mendenhall and Taku glaciers (Fig. 8c). The predictions suggest that glaciers in the south (e.g. Mendenhall, Taku, Lemon Creek) will have a SMB ~ 1.0 m w.e. a$^{-1}$ lower than elsewhere on the icefield. The glaciers with the lowest SMB by the end of the simulations are Mendenhall and Lemon Creek.

The surface mass balances shown in Fig. 8, from the mean of both models, lead to an increase in ELA of ~370 m by 2060,

with the highest increases (~400 m) seen at Field, Willison and Meade Glaciers in the north (Fig. S5). The projections show that the ELA of Lemon Creek remains above the elevation of its highest grid point. The AAR of glaciers across the icefield was found to decrease from 54% in the mean of the GFDL and CCSM historic simulations to 18% in future simulations (2031-2060), and to only 6% when considering the final decade of the simulations (2051-2060). The most rapid rise in ELA occurs during the first 10 years of the simulation, with the icefield wide AAR becoming ~10%. By the end of the future

simulations (2060), accumulation is confined to the highest elevations of the icefield.

## 4 Discussion

Here we discuss potential changes across the Juneau Icefield when comparing the historical and future simulations. We preface the following discussion with a note of caution. Our simulations of future SMB are projections, based on models which are imperfect representations of the real system. Furthermore, these projections are based upon RCP8.5, as this was

the scenario for which climate model simulations were available from Lader et al. (2020). It is worth reiterating that RCP8.5 is now seen as an unlikely scenario (Hausfather and Peters, 2020; Burgess et al., 2020), although the residence time of greenhouse gases in the atmosphere means that Juneau Icefield is likely already committed to the changes discussed below (Section 2.2).

### 4.1 Drivers of change across the Juneau Icefield

To investigate the potential drivers of the projected decrease in SMB (Fig. 8), we split the annual SMB from the GFDL simulation into its ablation and accumulation components (Fig. 9a). A significant positive correlation of 0.96 (p= 1.5e-08) between SMB and ablation over the period 2031-2060 (Fig. 9b), suggests that the interannual variability and negative trend in SMB is strongly linked to the variability of ablation processes. This increasing amount of ablation throughout the simulation is strongly linked to rising air temperature (Fig. 9c), which the GFDL model projects will be 3.5°C higher across

the icefield compared to 1980-2010 (Fig. 7a). Accumulation changes also have a strong correlation of 0.82 with SMB (Fig. 9a). In 2038 and 2059 there is high accumulation but no defined minima in ablation, leading to a peak in SMB (Fig. 9a). This contrasts with the historic simulation, where high snowfall years acted to reduce ablation.





The trigger for the projected future decrease in mass-balance of the Juneau Icefield is the changing climate. Warmer air temperatures will increase melt, and lack of snowfall will reduce the amount of accumulation. The combined effect of these

conditions means that across the icefield, for both the GFDL and CCSM projections, the ELA rises high on the plateau, meaning that the accumulation area shrinks to just 0.95% of the icefield. As this ELA rises, glaciers across the icefield are likely to lower in elevation, meaning more ice will be distributed to warmer climatic conditions, leading to an ice-elevation feedback (Böðvarssson, 1955). The hypsometry of Juneau Icefield, and likely other icefields around the world, means that it is particularly prone to the ice-elevation feedback (Hock and Huss, 2021; Bolibar et al. 2022) – small rises in ELA toward

the plateau mean a greater area of ice is in the ablation zone and undergoes thinning. Our projections suggest widespread thinning of outlet glaciers and negative mass balances on the plateau itself. These projections indicate that the plateau surface will no longer act as a source of snow and ice for outlet glaciers, but potentially undergo widespread thinning itself. In some places, the ice thickness of the plateau is greater than 600 m (Millan et al., 2022), meaning this non-linear mass balance feedback has a potentially large elevation range to operate over.

Rising ELAs across the icefield are likely to be exacerbated by an albedo-feedback (c.f. Johnson and Rupper, 2020). In the historic simulation, snowfall across large areas of the icefield increased the albedo of the icefield (Fig. 4c). Lower temperatures enabled larger regions of the glacier surface to be covered by snow for longer, effectively shielding the lower-albedo ice below. In the future projections (e.g. Fig. 7 d and h), snowfall covers a smaller area of the icefield for a shorter length of time, exposing the low albedo ice to more ablation. This temperature driven albedo effect is summarised in Fig. 10.

Increased temperature (Fig. 10b) leads to higher snowmelt rates (Fig. 10a), lowering albedo (Fig. 10c).

The thinning caused by the ice-elevation and ice-albedo feedbacks described above is augmented by several factors in the future simulations. The rise in air temperature impacts the ratio between solid (e.g. snow) and liquid (e.g. rain) precipitation. In the historic simulations, we found no correlation between air temperature and accumulation. This contrasts with the future projections, where there is a lower, but still significant, correlation of -0.48 (p=3.4e-13) between air temperature and

accumulation (Fig. 9e). Though annual precipitation reduces in future projections (Fig. 7 b and f), (Fig. 7 d and h): rising temperatures means less snowfall and more rainfall. This change in precipitation phase is most stark during autumn, when the Juneau Icefield usually receives the greatest amount of snowfall (Fig. 10d). In the evaluation and historic simulations, snowfall across Juneau occurs throughout the year, and starts to increase in August, with a peak in October (Fig. 10d). This autumn peak is due to the thermal contrast between the ocean and the land being at its highest around October, resulting in a

deeper Aleutian low, more frequent storms and large influxes of moisture (Wendler et al., 2016). In the future projections, daily snowfall does not increase until mid to late September, with an autumn peak only visible for the 2031-2035 period (Fig. 10c). This is despite an intensification of the Aleutian low in both models, due to the increasing ocean-land thermal contrast (Gan et al., 2017). Furthermore, the models project a lack of summer snowfall (Fig. 10d).

The importance of the albedo-melt feedback was highlighted in Johnson and Rupper (2020), who conducted idealised

experiments that showed that for a 1ºC increase in air temperature, 80% of the resultant melt is due to the albedo feedback. Our results suggest that this ice albedo feedback will play an increasingly important role over Juneau icefield in the future.



The hypsometry of the plateau and feedbacks on ice flow likely make the icefield more prone to, and exacerbate, these processes (Section 4.2). We also note that other mechanisms, not included in our model, could also act to decrease the albedo of the icefield. Large amounts of black carbon have reduced the icefield albedo (Nagorski et al., 2019). Concentrations of

black carbon and other deposits are expected to increase in the future, with a tourism-related rise in transport and increases in wildfires (Rain Coast Data, 2018; Kehrwald et al., 2020).

## 4.2 The potential response of glaciers across the Juneau Icefield

The interplay between ice flow processes, glacier setting, and glacier hypsometry will likely cause different outlet glaciers to respond differently to the projected dramatic rise in ELA. For many glaciers, especially those connected to the ice plateau,

the response is likely to be non-linear (Bolibar et al., 2022; Davies et al., in prep). Full exploration of these processes requires detailed ice-flow modelling similar to that of Ziemen et al. (2016). However, the results of our SMB modelling allow us to postulate on the potential fate of Juneau Icefield.

As well as triggering the ice-elevation and the ice-albedo feedbacks, the rise in temperatures also causes the ELA to rise above the elevation of many major icefalls. Davies et al. (2022), identified 13 major outlet glaciers at risk of disconnection

due to ice thinning intersecting their icefalls. By 2040, our model projects that all the major icefalls mapped by Davies et al. (2022) around the peripheries of the plateau will have a negative annual mass balance. Across the 2031-2060 time period, the multi-model mean of our future projections indicates a range of negative SMB at icefall locations, between -0.5 m w.e. a$^{-1}$ in the north to more than -2 m w.e. a$^{-1}$ in the south. The icefalls feeding Mendenhall and Herbert Glacier have a negative mean annual SMB of -1.8 m w.e. a$^{-1}$ in our projections (Fig. 11a). Of further concern are the East and West Twin glaciers,

with south-facing icefalls already observed to be thinning and narrowing in 2019 (Hugonnet et al., 2021; Davies et al., 2022). Our simulations of future SMB indicate that the mean annual SMB at these ice falls are -3.2 m w.e. a$^{-1}$ and -2.2 m w.e. a$^{-1}$ for the East and West Twin glacier icefalls, respectively (Fig. 11b). The high rates of thinning projected at these key locations suggest that glacier disconnection is likely if the projected temperatures are reached. Furthermore, the widespread thinning of the icefield that our SMB model projects will lead to the development of further icefall locations, as the ice thins over

steep topographic steps. Many of these icefalls, such as the one at West Twin Glacier (Fig. 11c), already display negative mass-balances. Thus, glacier disconnection at icefalls and subsequent rapid down-wasting may have already started and will potentially become more frequent in the future.

The hypsometry and catchment location of different glaciers means that the effects of this projected ELA rise are likely to alter between outlet glaciers. The reduction in accumulation area is most apparent in the southwest of the icefield, on the

windward slopes of the Taku Ridge. In our simulations of future SMB, this area becomes isolated from the other accumulation areas on the high elevation plateau, potentially shrinking outside of the catchment areas of glaciers such as Taku, Mendenhall and Norris. Most of the glaciers in the southwest are isolated from the high elevation plateau, and are thus are at risk of having increased retreat rates due to the projected minimal snow they will receive. The exception to this is Taku



glacier, which is connected to the plateau area. The over-deepened bed of Taku glacier is likely to continue to divert all snow
and ice from the plateau, away from its neighbouring glaciers.

The size, geographic setting, and terminus type may also play a role in determining the fate of Juneau Icefield's glaciers. Small, isolated mountain glaciers, such as Lemon Creek, lack attachment to the large catchment area of the main plateau, from which ice and snow may be sourced to delay retreat. The mass balance of Lemon Creek has been negative since 1953, leading to frontal retreat (Crisciteiello et al., 2010). Though Bolibar et al. (2022) suggest that the retreat of such glaciers to

higher elevations will mitigate losses caused by future warming, our historical simulations suggest that Lemon Creek glacier may already have an ELA close to or above its highest altitude and in the future projections, Lemon Creek has a strong negative SMB across its whole, gently sloping, surface. In such situations, the climatic forcing has likely outpaced any mitigating processes from topographic feedbacks with ice flow; instead of the ELA settling on a higher position on the glacier, the whole glacier surface is losing mass.

Conversely, tidewater glaciers are likely to have a different response under our SMB projections. The formerly tidewater Taku Glacier has historically been an outlier, as the only glacier to have advanced during the 20$^{th}$ Century whilst other glaciers on the icefield retreated (McNeil et al., 2020). Simple conceptual models suggest tidewater glaciers, such as Taku, undergo long retreat cycles as the terminus passes through their over-deepened basins. As the annual SMB of Taku becomes increasingly negative, as occurs in both of our future projections, the apparent resilience of Taku Glacier to climate change

may cease, with observed and modelled SMB negative between 2013 and 2019. This may trigger rapid retreat of Taku Glacier into its 40 km long over-deepened basin (Nolan et al., 1995) and re-instigate calving, increasing mass loss. Furthermore, if rapid retreat is triggered, then glacier terminus type is likely to rapidly alter through time. Tidewater glaciers or proglacial lake terminating glaciers may retreat beyond the water bodies they flow into, changing the dynamic processes influencing their flow. The start of this transition can already be observed at Mendenhall glacier, where retreat has resulted

in half of the terminus being situated on land above the proglacial lake (Fig. 11d).

Higher air temperatures across the icefield, which reduce the amount of precipitation that falls as snow, cause a decrease in the depth and spatial area of the winter snowpack. This causes a lower albedo during the summer, with each subsequent 5-year period shown in Fig. 10d to decrease toward the bare ice albedo of 0.45 in COSIPY. One projected response of the icefield to these drivers is a corresponding projected change in melt season. If we define the onset of the melt season as being the date on which there was more than 1 mm day$^{-1}$ of melt at greater than 5% of icefield grid-points, then there is an

increased duration of melt season of more than a month during the last five years of the future projections (Fig. 10a). This increase in melt season length, is complimented by an increase in the peak surface meltwater production of the icefield, with the 5-day maximum increasing by 35 m w.e. day$^{-1}$ when future projections are compared to the simulations of the historic period. This comparison reveals an increase in total annual surface meltwater production of 5542 m w.e. a$^{-1}$, an increase of

139%. A potential impact of this is that increased surface meltwater may be more likely to drain to the bed of the glacier, causing brief accelerations of ice flow (e.g. Iken and Bindschadler, 1986; Burgess et al., 2013). This effect could be even more prominent due to the peak in surface meltwater occurring early in the season, before the development of efficient sub-

glacial drainage systems. Such spring glacier speed-up events have been recorded previously (e.g. Anderson et al., 1999; Macgregor et al., 2005; Bartholomaus et al., 2008). If this is the case, accelerated ice flow will deliver more ice towards the
lower elevation and warmer temperatures.

## 5 Conclusions

High resolution dynamically downscaled climate models (Lader et al., 2020) have enabled us to simulate the historical and projected future SMB of Juneau Icefield using the COSIPY model. Tuning of the model to the rich empirical record collected by the Juneau Icefield Research Project enabled us to accurately simulate the pattern of past changes and provides
confidence in future projections. This highlights the value of such long-term monitoring programmes. Under RCP8.5 projections, our modelling suggests that the mass balance across Juneau Icefield is set to become increasingly negative in the middle of the 21$^{st}$ century, with a dramatic rise in ELA and reduction of the accumulation area. For the period 2031-2060, a multi-model mean of -1.52 m w.e. a$^{-1}$ is projected. This is attributed to an icefield-wide increase in air temperature, which causes increased snowmelt and a higher percentage of precipitation to fall as rain rather than snow. Reduction of snow cover
in the model leads to longer and more extensive exposure of lower albedo ice, leading to an albedo-induced melt feedback. Negative mass balances are likely to spread across the plateau, and at icefalls ice thinning is likely to promote glacier disconnections. These stark projections of future mass balance are likely to lead to numerous feedbacks which augment future ice losses from the Juneau Icefield. Future work should consider ice-flow feedbacks, and the timescales over which similar processes are likely to occur on other plateau-icefields and ice caps globally.

**Data availability**

The output from COSIPY will be deposited upon acceptance of the manuscript.

**Author contribution**

RNI led the data processing, modelling, and analysis, under the supervision of JCE and JMJ. BJD provided additional data on glacier extent and field photos. JCE led the preparation of the manuscript, with input from all authors.

**Acknowledgements**

RI acknowledges the support of the Karen Reed Memorial Award (Charity no. 1085619-1) and the University of Sheffield Postgraduate Scholarship, which funded him to study a MSc(Res) in Polar and Alpine Change during which this work was undertaken. JCE acknowledges support from a NERC independent fellowship (NE/R014574/1).



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



# Figures




**Figure 1: Location of the Juneau Icefield and key glaciers mentioned within the text.**





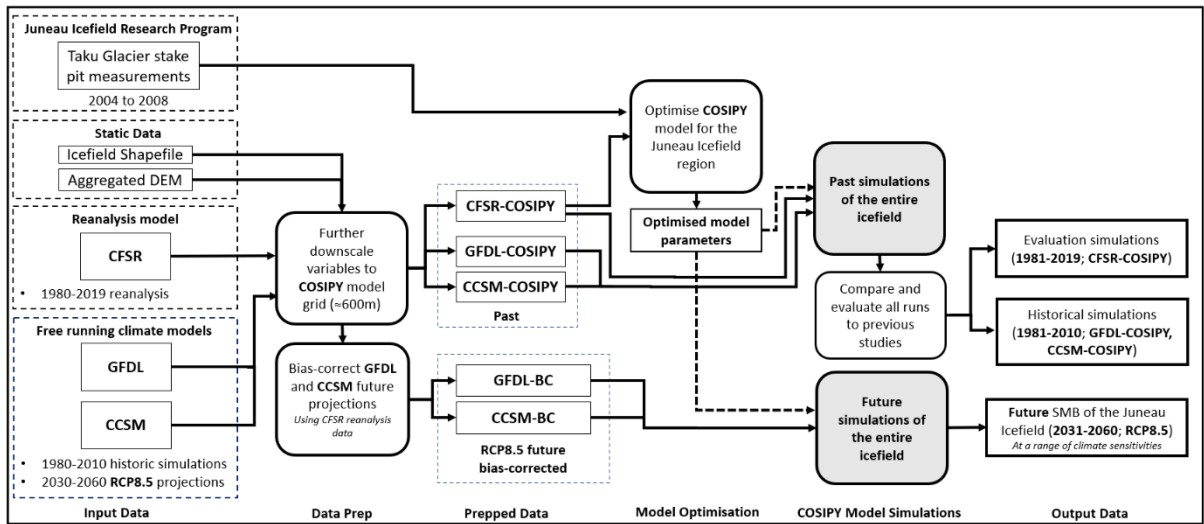

**Figure 2: Overview of the workflow of this study. The schematic highlights which of the three climate model outputs from Lader et al. (2020) are used in each experiment.**



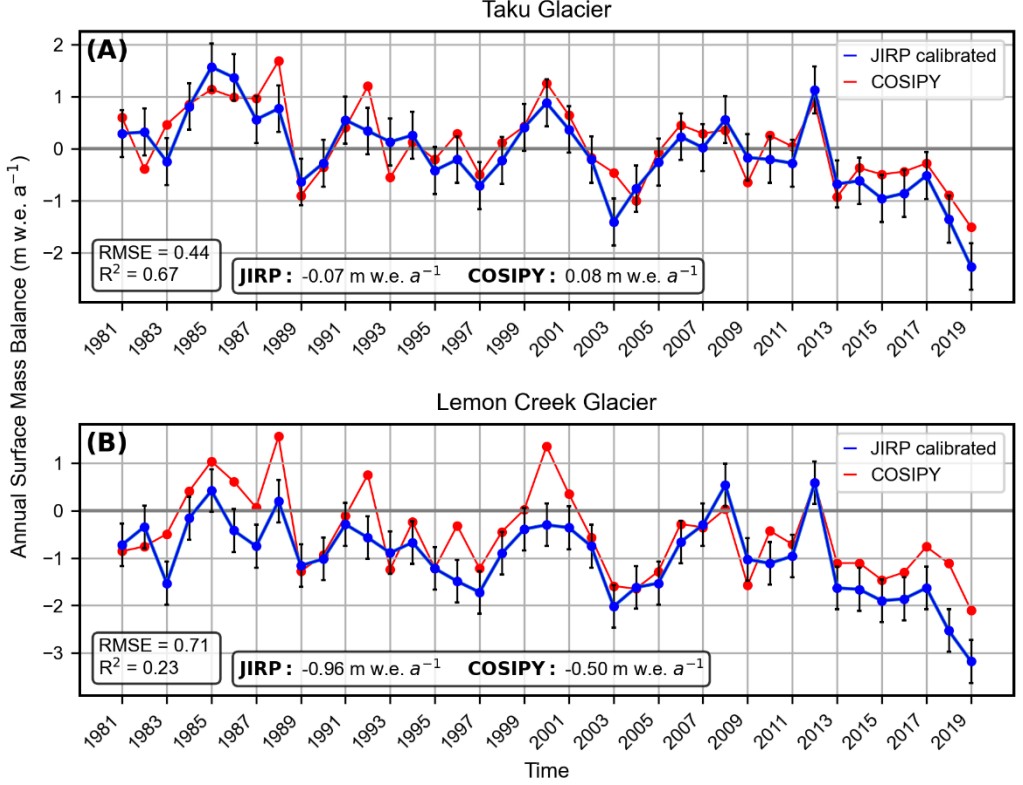

**Figure 3. Comparison between the COSIPY simulated SMB (red) and the calibrated JIRP SMB (blue) for a) Taku Glacier and b) Lemon Creek Glacier, between 1981 and 2019 using the CFSR-reanalysis data. The bottom left box on each subplot displays the RMSE and R2 between the COSIPY model and the JIRP SMB data.**





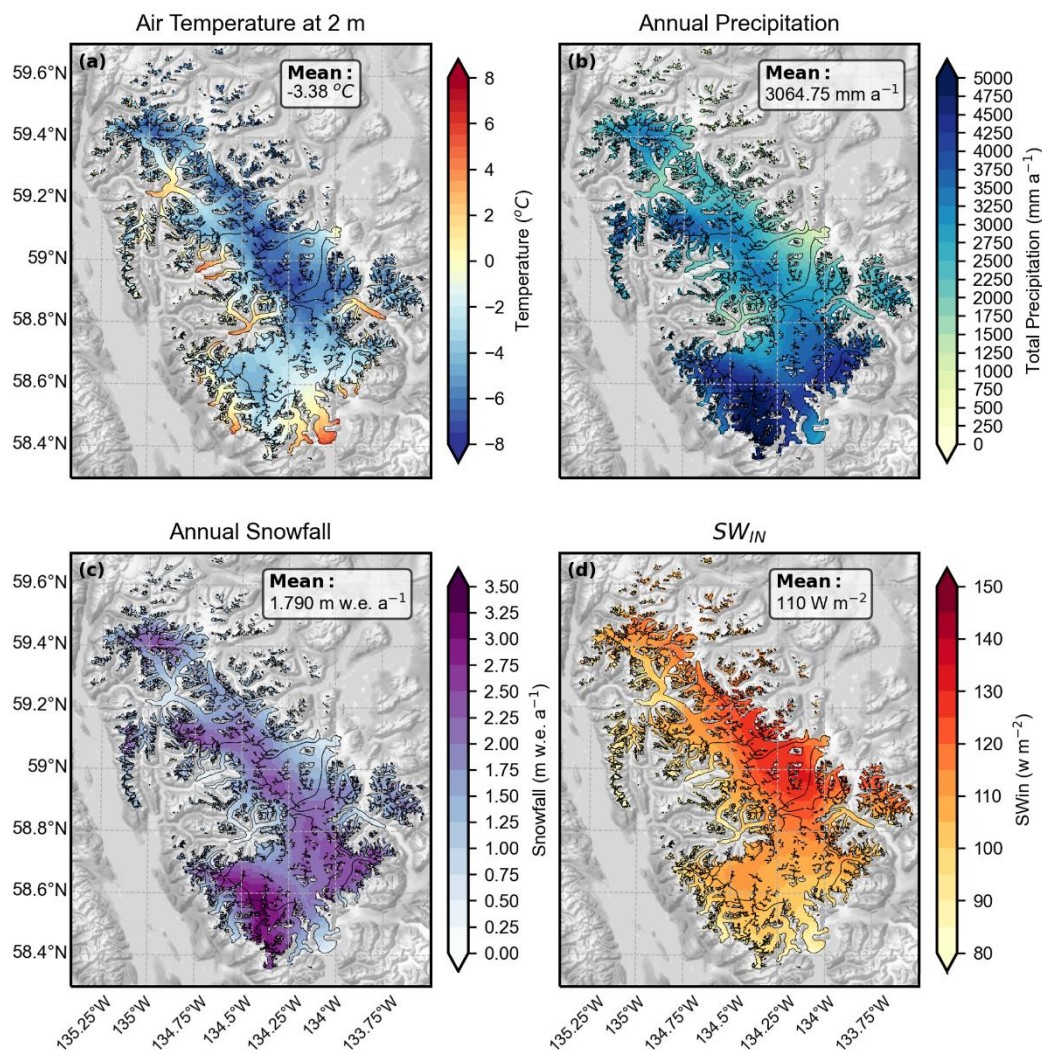

**Figure 4. Climate average for Juneau Icefield from downscaled CFSR data. a) Mean daily air temperature at 2 m, b) mean total annual precipitation, c) mean annual snowfall, d) mean incoming shortwave radiation.**




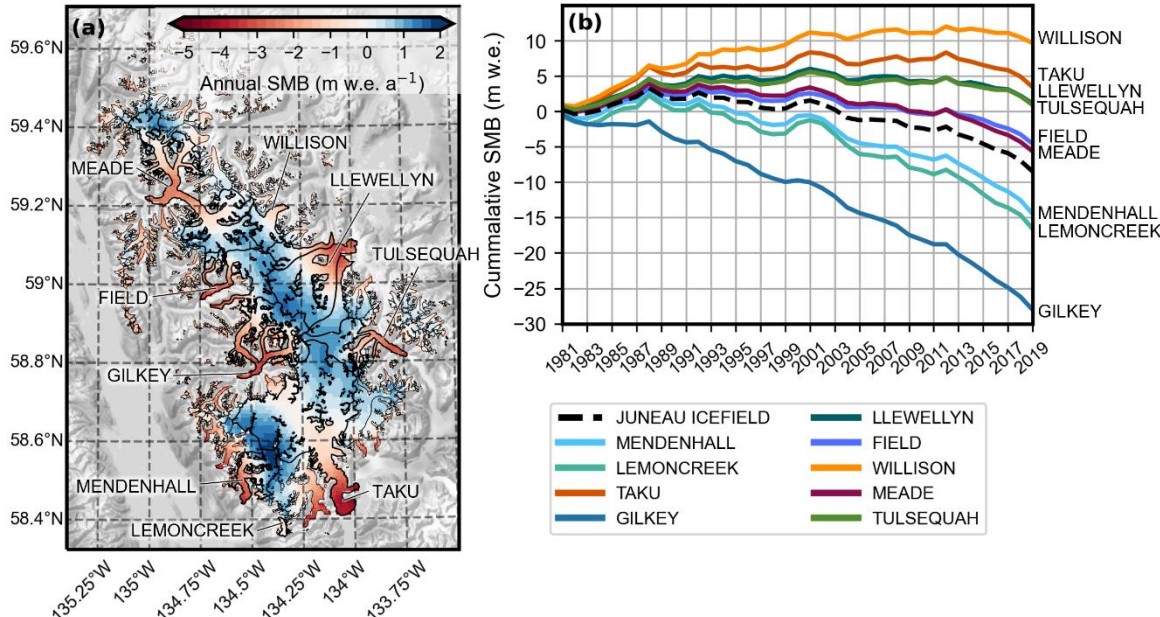

**Figure 5. Surface mass balance estimates from CFSR-COSIPY during the historic period (1981-2019). a) The mean annual surface mass balance. b) The annual cumulative surface mass balance of select glaciers across Juneau Icefield.**




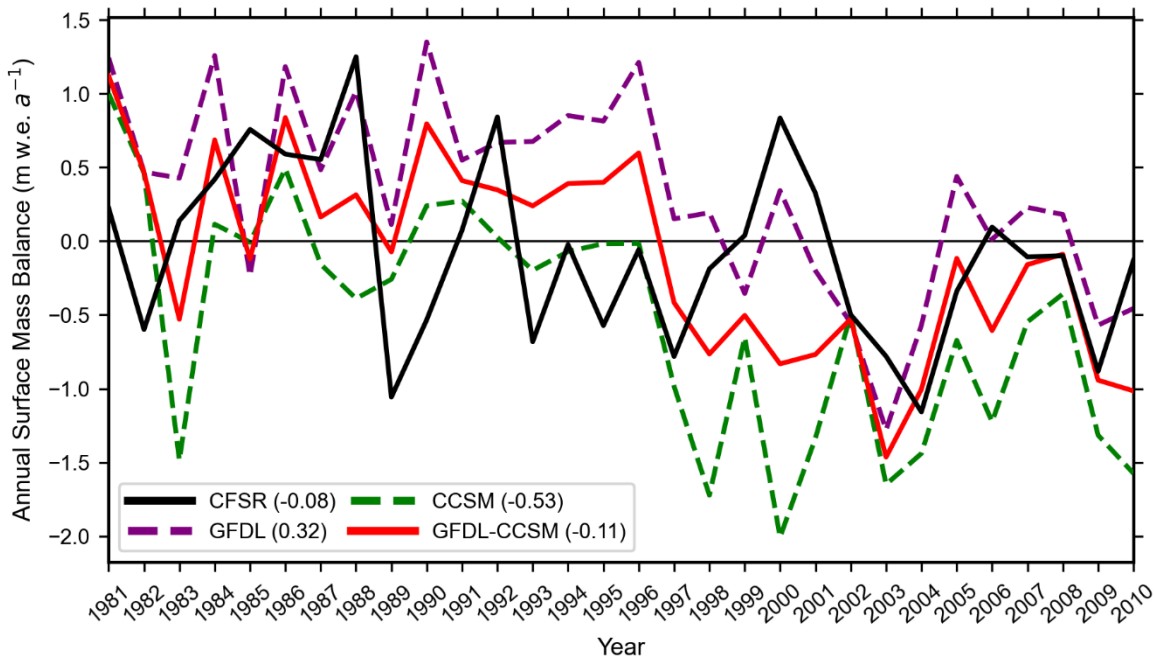

**Figure 6. Time series of past (1981-2010) SMB across Juneau Icefield from three different climate simulations. The average annual SMB of the whole period noted next to each model in the legend, with units of (m w.e. a-1).**


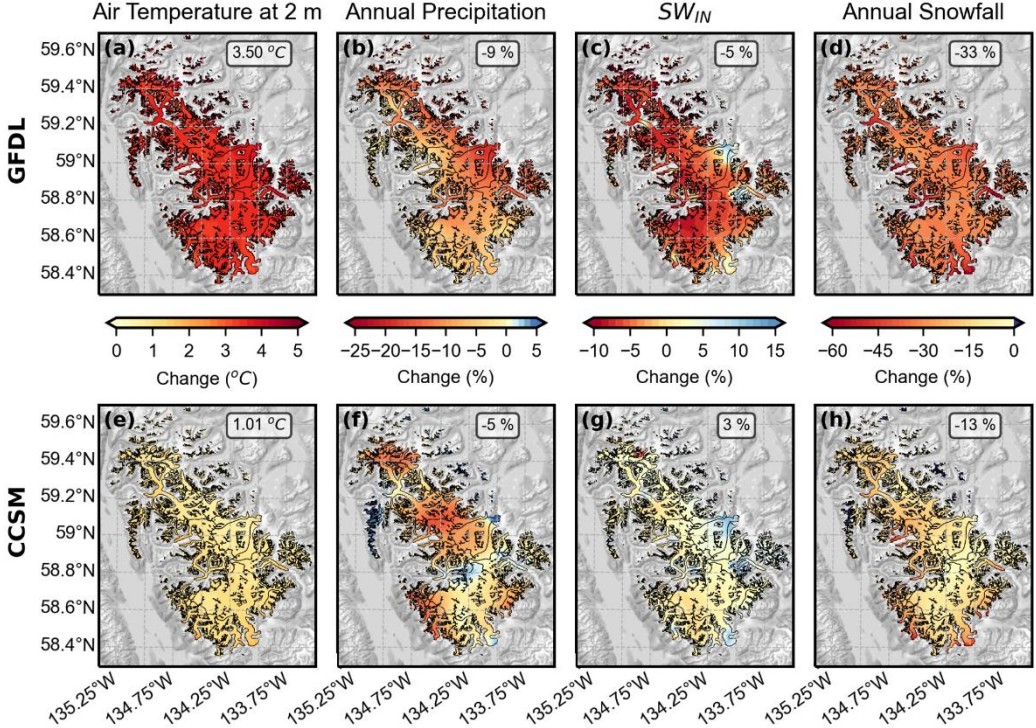


**Figure 7. Changes in the future climate averages for the Juneau Icefield domain between the future (2031-2060; RCP8.5) and historic (1981-2010) means. (a) change in daily 2 m air temperature, (b) percentage change in total annual precipitation, (c) percentage change in daily incoming shortwave radiation and (d) percentage change in annual snowfall for the GFDL-COSIPY data. (e -h) same as (a - d) but for the CCSM-COSIPY data. Additional text**
**boxes on each plot show the average change for the whole icefield domain.**



**Figure 8. Projections of future SMB of Juneau Icefield (2031-2060, RCP8.5). a) Time series of future average SMB across the icefield for each model. The average annual SMB of the whole period is noted next to each model in the legend, units are m w.e. a-1. Regional projections of specific total mass balance from Hock et al. (2019) are displayed in grey. b) The spatial distribution of the mean annual SMB from the mean of GFDL and CCSM. c) The change in mean annual SMB compared to the GFDL and CCSM simulated mean from the historic period.**





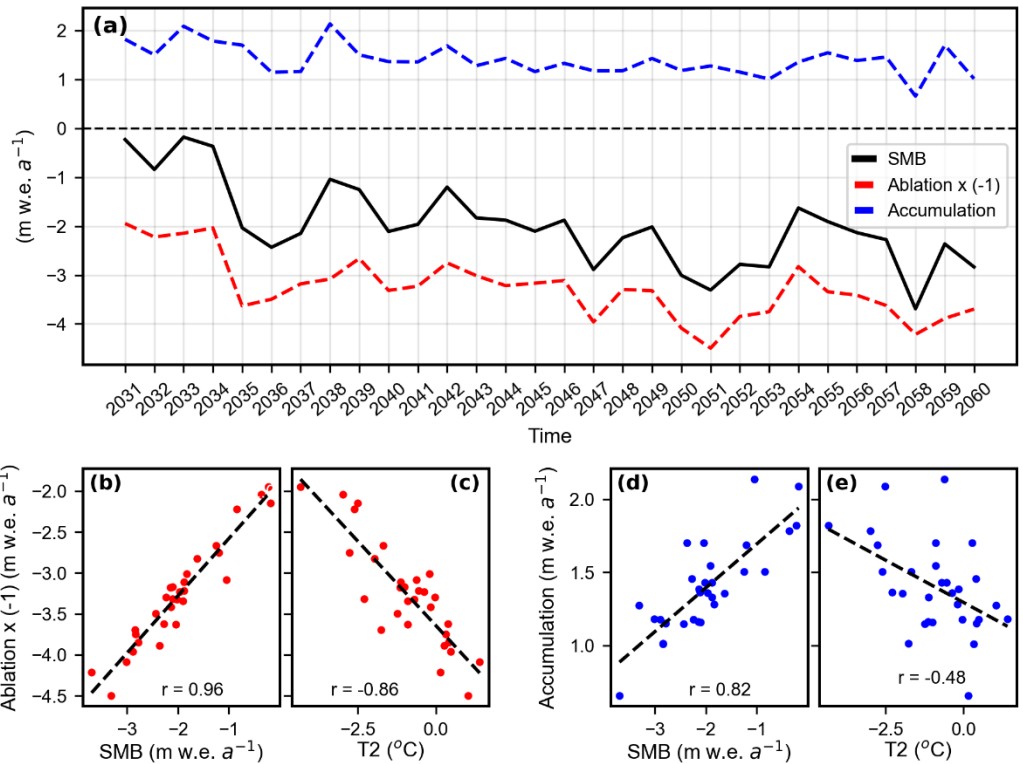

**Figure 9. Potential drivers of future change across Juneau Icefield. a) Time series for the GFDL future simulation, separating SMB, ablation, and accumulation. Note that the sign of ablation has been swapped, so that a decrease on the graph represents more ablation. b-e) Scatter plots showing the relationships between different variables and the ablation and accumulation across the icefield. The Pearson correlation coefficient (r) is shown for each.**



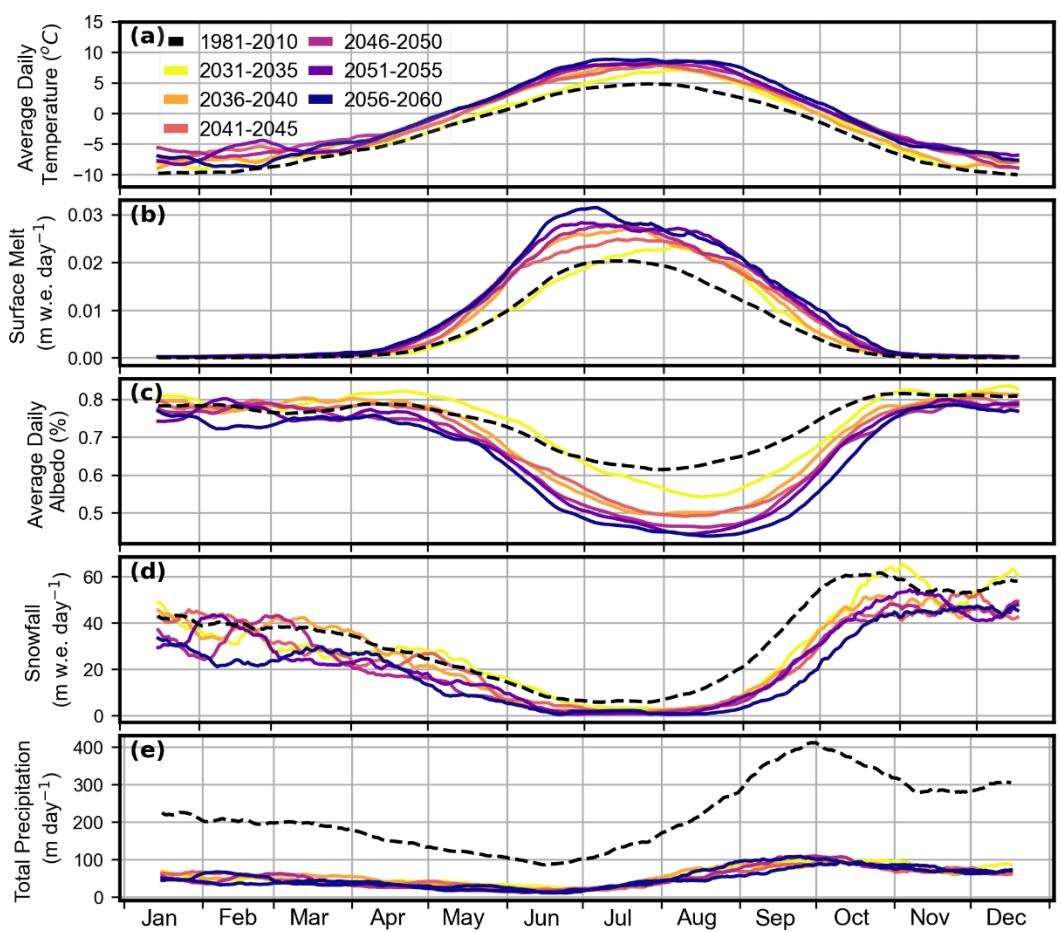

**Figure 10. The historical and projected changing seasonal cycle across Juneau icefield from the mean of the GFDL and CCSM simulations. a) air temperature, b) surface melt, c) albedo, d) total icefield-wide snowfall and e) total icefield-wide rainfall. The mean of all modelled cells is shown, and a 30-day running mean has been applied to all fields.**





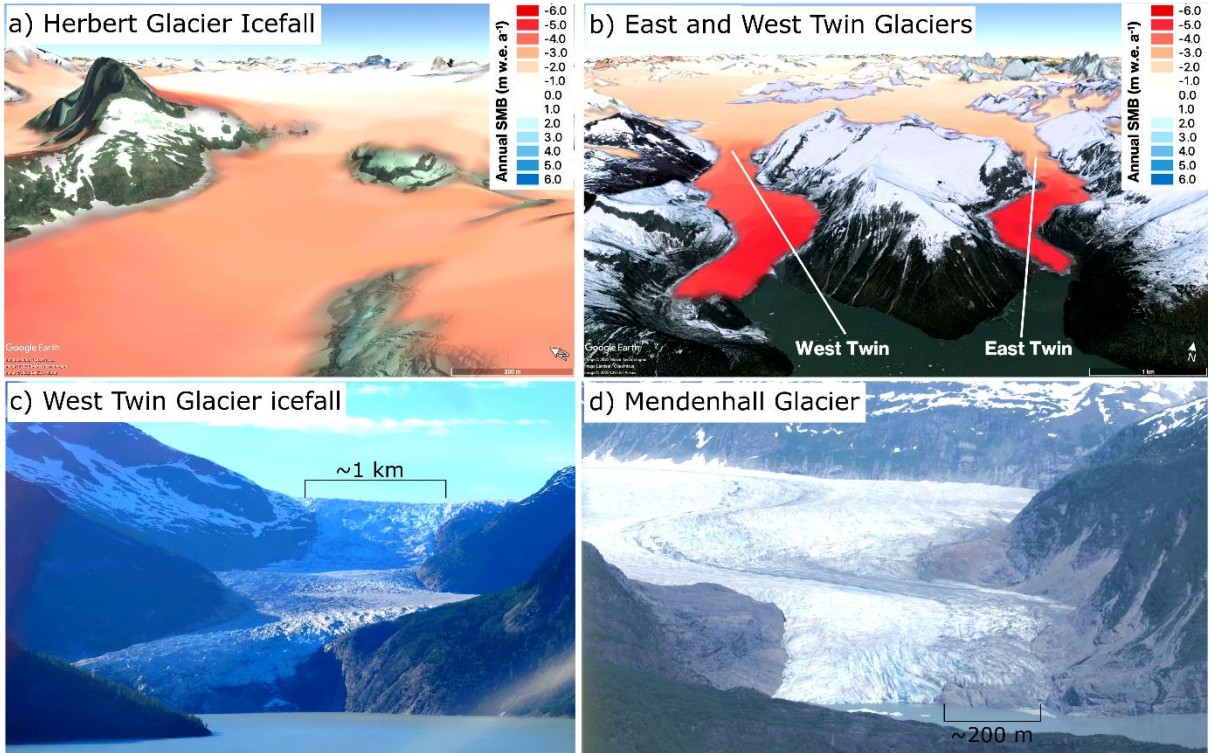

**Figure 11. Potential future and ongoing changes to outlet glaciers of Juneau icefield. a) and b) Mean annual SMB for 2050-2060 from the GFDL and CCSM simulations superimposed and interpolated on © Google-Earth imagery, for key icefalls on a) Herbert Glacier and b) West and East Twin glaciers. Here, the annual mass balance is predicted to be highly negative. These areas are thus likely locations of future glacier detachment. c) Photo of West Twin Glacier icefall (July 2022), a location already undergoing thinning. d) The terminus of Mendenhall Glacier (July 2022). Note how the front has already partially receded from its proglacial lake.**

665





Table 1. Optimised model parameters. Note that the initial testing range is adapted from those tested in Blau et al. (2021).

| Model Parameter | Range | Optimized value |
|---|---|---|
| Albedo of fresh snow | 0.8 – 0.9 | 0.87 |
| Albedo of firn | 0.46 – 0.79 | 0.5 |
| Albedo of bare ice | 0.3 – 0.45 | 0.4 |
| Effect of ageing on fresh snow albedo (days) | 1 – 21 | 4.3 |
| Effect of snow depth on albedo (cm) | 1 – 30 | 10.7 |
| Surface roughness length for fresh snow (mm) | 0.24 – 0.5 | 0.5 |
| Surface roughness length for firn (mm) | 0.7 – 2.7 | 1.8 |
| Surface roughness length for bare ice (mm) | 2.4 – 5.0 | 2.5 |
| Surface emission coefficient | 0.97 – 1.0 | 0.975 |
| Multiplication factor for total precipitation | 0.7 – 1.1 | 0.74 |

670





Table 2. Comparison of the results of this study to previous estimates of specific mass balance of the Juneau Icefield. Note the different periods of study. The area of the icefield is not listed for Ziemen et al. (2016) as this evolves with modelled ice dynamics.

| Study | Period | Area (km$^2$) | Specific Mass Balance (m w.e. a$^{-1}$) |
|---|---|---|---|
| Larsen et al. (2007) | 1948/1982/1987-2000 | 3410 | -0.62 |
| Berthier et al. (2010) | 1948/1982/1987-2006 | 2960 | -0.53 +/- 0.15 |
| Melkonian et al. (2014) | 2000-2010 | 3830 | -0.13 +/- 0.12 |
| Ziemen et al. (2016) | 1971-2010 | * | -0.33 |
| Berthier et al. (2018) | 2000-2016 | 3398 | -0.68 +/- 0.15 |
| *This study (CFSR) whole time period* | *1981-2019* | *3818* | *-0.33* |
| *This study (CFSR) for comparison to Berthier et al. (2018)* | *2000-2016* | *3818* | *-0.55* |

675