# Peer review of "Surface mass balance modelling of the Juneau Icefield highlights the potential for rapid ice loss by the mid-21st century"

_The Cryosphere, 2023_

## Referee Comment (RC2)

Dear Ryan and co-authors, Dear Harry,

Please find below my review on the article "Surface mass balance modelling of the Juneau Icefield highlights the potential for rapid ice loss by the mid-21st century".

**Summary**

The authors use the COSIPY energy and mass balance model to improve the surface energy balance (SEB) on the Juneau Icefield. The model is used to simulate historical and future mass changes. COSIPY is driven with dynamically downscaled historical and the RCP8.5 climate scenario and calibrated using long-term in-situ observations. Mean mass changes over the period 2031-2060 are finally compared to the historical period 1981-2010. The authors conclude that the simulated mass loss is largely caused by reduced snow precipitation leading to plateau-wide glacier thinning.

**General comment**

Although the work does not substantially change our scientific understanding of the Juneau Icefield, it supports and strengthen previous findings. The utilization of numerical energy and mass balance models represents a significant step toward consolidating existing knowledge. The aim of the work is clearly formulated, and the methodology is well chosen to improve the surface mass balance estimates on the Juneau Icefield. The findings are well supported with figures and illustrations and appropriately discussed in relation to previous studies. In summary, the authors present their results and conclusions in a clear, concise, and well-structured way.

However, there are a few important aspects that should be considered before the article can be published:

**(1) Uncertainties** play an important role in the analysis of climate scenarios. Nowadays, the inclusion of uncertainty estimates is considered good scientific practice. Determining these uncertainties is not always easy and the determination is subjective. The important aspect of uncertainties is something that has been completely neglected in this study. The results are based on only two models and one scenario (RCP8.5). If the entire CMIP5 ensemble were considered, the uncertainties in the SMB estimates would certainly be correspondingly large. In addition to the ensembles, model uncertainties from COSIPY should also be considered. It is difficult to make specific recommendations for dealing with these uncertainties. However, you should include as many CMIP5 members as possible in your study (Lader et al. (2020) certainly included several more ensemble members) and determine the uncertainties of the COSIPY model parameters using, for example, Monte Carlo runs. At the end, an uncertainty range should be given for all SMB simulations.

**(2) Drivers of change across the Juneau Icefield.** The increase in the equilibrium line altitude (ELA) into the higher regions of the plateau and the ice-elevation feedback are mentioned as key mechanisms that accelerate the melting of the plateau. So far, the discussion has been very vague, and there is a lack of reliable facts. It would be highly interesting to quantify the effect of the ice-elevation feedback and determine its impact on the surface mass balance (SMB) trend. The same applies to the ice-albedo feedback, which is equally exciting. To quantify the contribution of these feedback mechanisms, further simulations need to be conducted, where the digital

elevation model (DEM) is updated annually in the model. I can only encourage you to quantify these feedbacks, as this would greatly increase the importance and visibility of your work in the scientific community. Honestly, I would put the focus of this paper on the feedbacks rather than writing another "mass balance of" paper.

**(3) The potential response of glaciers across the Juneau Icefield**
The analysis of the potential response of glaciers could be further improved. As correctly mentioned, factors such as size, setting, etc., play a role. Fundamentally, glacier dynamics should be considered to account for the influence of mass changes on glacier length. Only by doing so, one can make a reliable statement regarding the response of glaciers. One approach could be to drive the OGGM model using the mass balance simulations from COSIPY. That is indeed a somewhat greater effort, but only in this way can one assess or quantify the effect of changes in the climate signal on the outlet glaciers.

**Specific comments**

**Input data to COSIPY**

P5L150: It would be good to mention briefly why the two models rank in the top five of all CMIP5 models for Alaska.

P5L155: It is comprehensible that only the RCP8.5 scenario was used here due to the available simulations of Lader et al. (2020). However, I do not share the opinion that the choice of scenarios is irrelevant for the selected period. As can be seen from the previous paragraph, the two models have very different climate sensitivities, which is especially noticeable in more extreme scenarios. When interpreting the RCP8.5 model results, the different climate sensitivities of the models must be considered. It would be also nice to see what the differences between RCP8.5 and RCP3.7 are in this region.

P5L172: Bias corrections usually assume that the biases at the quantiles do not change over time (stationarity). Thus, the correction itself leads to further errors in the time series, or does it not?

**Model optimisation**

P6L189: Why were only 100 random samples generated for parameter optimization? Typically, several hundred or thousand parameter combinations are used.

**Historical simulations of SMB from climate models (1980-2010)**

P8L256: What do you mean exactly with statistically similar?

**Future SMB of the Juneau Icefield (2031-2060, RCP8.5)**

P9L285: Why do the two SMB time series show a very similar trend even though the climate trends and climate sensitivities are so different?

P9L287: How would the result change to that of Hock et al. (2019) if the change in glacier hypsometry were considered? Wouldn't the results of the two studies then be further apart?

P11L322: Why has this physical logic reversed, and snowfall no longer reduces ablation? Is there an explanation for this?

---

## Author Comment (AC1)

**TC-2023-33**

**Response to reviewers**

Reviewers comment; Author response; Proposed Manuscript Change for resubmission

We would like to thank Andrew Bliss for his comment, and Anonymous Referee #1 and Tobias Sauter (Referee #2) for their reviews. We have considered all the provided comments and respond below. We are pleased that both reviewers highlight the novelty of our work, and note that their comments will lead to improvements in the manuscript.

**CC-1**

How do your historical results compare to Young et al 2021?

https://agupubs.onlinelibrary.wiley.com/doi/full/10.1029/2020WR027404

Thank you for highlighting this publication, which we previously missed. Our evaluation simulations have a slightly less negative mass balance rate than reported in Young et al. 2021. This likely results from differences in domain, resolution of reanalysis data used to force the model, and differences in processes included in the models.

We will include the estimates of mass balance from Young et al. (2021) in Table 2, and highlight these differences in section 3.1.

**RC-1**

Ryan et al. give new estimates about the surface mass balance of the Juneau Icefield at the north American east coast. The use the Coupled Snowpack and Ice surface energy and mass-balance model in Python to generate a data set of the surface mass and energy balance in present day and future. For calibration and validation of COSIPY, they used input data from the CFSR Reanalysis, for future projections in the period 2031 to 2060 they used input data from output of RCP8.5 scenarios of GFDL-CM3 and NCAR-CCSM4 (CMIP5 models). The validation showed promising results. Future projections indicated largescale ablation and the risk of glaciers which loose connection to the ice source region. The COSIPY output for future projections gives insights in possible outcomes assuming different climate sensitivities.

The manuscript fills a gap on the mass balance of the Juneau Icefield. This was done by using CMIP5 models as input to COSIPY. Therefore, it is a valuable addition to science. The dataset can give estimates of the future development of the ice sheet and invites for novel research.

We thank this reviewer for their thorough review, and for highlighting the novel contribution of this manuscript.

*Major comments*

You applied bias correction on the downscaled model input. How did the bias correction improve the input data?

The bias correction utilised the CFSR reanalysis data to remove any inherent bias in the future projections of each model (GFDL and CCSM), through comparing the historic and reanalysis simulations. The differences between the un-corrected and corrected versions are minimal. For GFDL it generally acted to decrease the annual SMB and increase it for the CCSM projections.

This will be clarified on L173.

Can you give further explanation for the decrease in total precipitation (Fig 10e)?

We have updated figure 10 to show the mean precipitation and snowfall over the icefield rather than the sum. Part of the decrease in precipitation was due to future simulations having more "no data" values, and thus an oversight on our behalf. Given that the future dataset is only 30 years, the changes in precipitation are likely to reflect interannual variability, rather than a signal of change.

We have updated Figure 10 as below, and we will update the manuscript accordingly, explaining the apparent change in precipitation is likely noise.

[Figure]

In Fig 10 there is a strong decrease in total precipitation but only small decrease in snowfall. Would this lead to more precipitation falling as snow in future? Which implication does that have? Is it removed by increased snowmelt?

Please see the above comment. The average mean total precipitation of the grid cells does increase during the wettest part of the year (September/ October), however due to the warmer air temperatures the majority of this falls as rain.

We will carefully clarify the discussion of this (around lines 281-282).

In some figures the units are not matching.

We will carefully check units throughout the manuscript before resubmission.

*Minor comments:*

Why did you choose GFDL-CM3 and NCAR-CCSM4? Did you evaluate other models as well?

The dynamical downscaling of the models was performed in Lader et al. (2020). They chose these models as they were amongst the top performing out of all CMIP5 members for Alaska and additionally have been shown to be at opposing ends of climate sensitivity (Walsh et al. 2018, Flato et al. 2013).

Can you specify which of the outlet glaciers are marine-terminating or have floating glacier tongues?

 Taku Glacier is the only marine-terminating glacier on the icefield. A few of other glaciers do terminate into pro-glacial lakes (e.g. Gilkey glacier).

We will add a sentence to clarify this.

L. 124: roughness length decline?

For the snowpack, roughness length increases linearly with time from snow to firn (Sauter et al. 2020).

L. 131: "… shapefile of the region of interest …" can be rephrased. The shapefile defines the grid points on which COSIPY is applied (glacier mask)

Agreed. This will be changed for resubmission.

L. 218 & Fig 4: Is the unit correct? 3060 mm a-1 seems high.

Units are correct. Similar annual totals were reported in Lader et al. (2020) for the whole of Southeast Alaska and Wendler et al. (2017).

L. 280: predicted -> projected to?

Will amend accordingly.

L. 292: towards to end -> towards the end?

Will amend accordingly.

L. 296: prediction -> projection?

Will amend accordingly.

L. 326-328: the sentence is unclear. Can you please rephrase it?

We will change this to: "With the rising ELA the icefield is likely to thin, lowering its elevation to warmer climatic conditions leading to an ice-elevation feedback (Böðvarssson, 1955)."

L. 340: Please check the panels in the figure reference?

Will amend accordingly.

L. 433: multi-model mean SMB

Will amend accordingly.

Fig 10: Are snowmelt and snowfall in the same unit (mm w.eq. vs. m w.eq.)?

Fig. 10 now shows the mean snowfall. Both surface melt and snowfall are in m w.e.

This is incorporated into the change in Figure 10.

**RC-2**

Dear Ryan and co-authors, Dear Harry,

Please find below my review on the article "Surface mass balance modelling of the Juneau Icefield highlights the potential for rapid ice loss by the mid-21st century".

*Summary*

The authors use the COSIPY energy and mass balance model to improve the surface energy balance (SEB) on the Juneau Icefield. The model is used to simulate historical and future mass changes. COSIPY is driven with dynamically downscaled historical and the RCP8.5 climate scenario and calibrated using long-term in-situ observations. Mean mass changes over the period 2031-2060 are finally compared to the historical period 1981-2010. The authors conclude that the simulated mass loss is largely caused by reduced snow precipitation leading to plateau-wide glacier thinning.

*General comment*

Although the work does not substantially change our scientific understanding of the Juneau Icefield, it supports and strengthen previous findings. The utilization of numerical energy and mass balance models represents a significant step toward consolidating existing knowledge. The aim of the work is clearly formulated, and the methodology is well chosen to improve the surface mass balance estimates on the Juneau Icefield. The findings are well supported with figures and illustrations and appropriately discussed in relation to previous studies. In summary, the authors present their results and conclusions in a clear, concise, and well-structured way.

We take this opportunity to again thank Referee #2 (Tobias Sauter) for their constructive review. We are pleased that they recognise the clarity of our manuscript.

However, there are a few important aspects that should be considered before the article can be published:

    (1)  Uncertainties play an important role in the analysis of climate scenarios. Nowadays, the inclusion of uncertainty estimates is considered good scientific practice. Determining these uncertainties is not always easy and the determination is subjective. The important aspect of uncertainties is something that has been

completely neglected in this study. The results are based on only two models and one scenario (RCP8.5). If the entire CMIP5 ensemble were considered, the uncertainties in the SMB estimates would certainly be correspondingly large. In addition to the ensembles, model uncertainties from COSIPY should also be considered. It is difficult to make specific recommendations for dealing with these uncertainties. However, you should include as many CMIP5 members as possible in your study (Lader et al. (2020) certainly included several more ensemble members) and determine the uncertainties of the COSIPY model parameters using, for example, Monte Carlo runs. At the end, an uncertainty range should be given for all SMB simulations.

Overall, we agree with the point regarding the importance of uncertainties. However, we disagree that uncertainties have been neglected in this study.

First, we justify our choice of model simulations, and how they influence uncertainty reporting:

- The choice of two CMIP5 members is a pragmatic one. The referee is incorrect in their statement that Lader et al. (2020) included more CMIP5 members, only two were dynamically downscaled. Furthermore, only RCP8.5 was used by Lader et al. (2020). Thus, these are the only two dynamically downscaled RCM simulations available to us.
- Despite being a pragmatic choice for us, these two model members were carefully chosen by Lader et al. (2020). They reflect high and low sensitivities to climate (which we repeatedly state throughout the manuscript; L152, L153, L285. Thus, these two model members likely encapsulate a large amount of the model uncertainty from CMIP5 members (akin to a minimum and a maximum estimate). We report both of these values throughout the manuscript, thus encapsulating as much uncertainty from downscaled CMIP5.
- The RCP8.5 simulations are the only available dynamically downscaled simulations we have access to. As we state on lines 155-161, refer the reader back to on line 270, and reiterate on lines 310-313, the residence times of greenhouse gases means that we are likely already committed to RCP8.5 on the timescales in question, meaning alternative scenarios are unlikely to differ from RCP8.5.

Turning to the potential uncertainties arising from the COSIPY parameters. One could envisage reporting projections from all potential parameter combinations (i.e. run simulations using parameter combinations from our Latin Hypercube sample). However, many of these combinations may produce garbage simulations. Instead, we choose to run only our "best estimate" simulation – the one that best fits our data on the past behaviour. Interestingly, this somewhat mirrors a debate in climate modelling: should we have a one vote one model system, or weight models we know to perform better more highly (Collins et al., 2017; Knutti et al., 2017)?

Furthermore, previous COSIPY papers do not report perturbed parameter ensemble experiments (e.g. Sauter et al., 2020; Blau et al., 2021). In the absence of an agreed method for reporting such uncertainties (e.g. weighted, optimum run, all parameters), we prefer to stick to our current optimised method.

(2) Drivers of change across the Juneau Icefield. The increase in the equilibrium line altitude (ELA) into the higher regions of the plateau and the ice-elevation feedback are mentioned as key mechanisms that accelerate the melting of the plateau. So far, the discussion has been very vague, and there is a lack of reliable facts. It would be highly interesting to quantify the effect of the ice-elevation feedback and determine its impact on the surface mass balance (SMB) trend. The same applies to the ice albedo feedback, which is equally exciting. To quantify the contribution of these feedback mechanisms, further simulations need to be conducted, where the digital elevation model (DEM) is updated annually in the model. I can only encourage you to quantify these feedbacks, as this would greatly increase the importance and visibility of your work in the scientific community. Honestly, I would put the focus of this paper on the feedbacks rather than writing another "mass balance of" paper.

(3) The potential response of glaciers across the Juneau Icefield The analysis of the potential response of glaciers could be further improved. As correctly mentioned, factors such as size, setting, etc., play a role. Fundamentally, glacier dynamics should be considered to account for the influence of mass changes on glacier length. Only by doing so, one can make a reliable statement regarding the response of glaciers. One approach could be to drive the OGGM model using the mass balance simulations from COSIPY. That is indeed a somewhat greater effort, but only in this way can one assess or quantify the effect of changes in the climate signal on the outlet glaciers.

We address both of these points here. Both suggestions are valid, but require potentially years of further work. Essentially both would require coupling between a mass balance model and an ice flow model. To our knowledge, COSIPY is yet to be coupled to an ice flow model (e.g. OGGM). Doing so is likely a large undertaking requiring technical expertise (probably postdoctoral researchers), and achieving such coupling is likely a separate paper in itself. Nevertheless, these are interesting avenues of future research that we would enjoy pursuing, perhaps with the guidance of the reviewer.

In the absence of such a coupled model setup, addressing these points is beyond our current capabilities. Hence, why our discussion (not results or methods) angles towards the speculative rather than the specific.

*Specific comments*

Input data to COSIPY

P5L150: It would be good to mention briefly why the two models rank in the top five of all CMIP5 models for Alaska.

This will be expanded upon in resubmission.

P5L155: It is comprehensible that only the RCP8.5 scenario was used here due to the available simulations of Lader et al. (2020). However, I do not share the opinion that the choice of scenarios is irrelevant for the selected period. As can be seen from the previous paragraph, the two models have very different climate sensitivities, which is especially noticeable in more extreme scenarios. When interpreting the RCP8.5 model results, the different climate sensitivities of the models must be considered. It would be also nice to see what the differences between RCP8.5 and RCP3.7 are in this region.

Unfortunately, the RCP3.7 scenario is unavailable as a dynamically downscaled experiment of the region. We take the point that the different climate sensitivities combined with different pathways may produce different temperature ranges. Future experiments could follow this line of enquiry.

We will raise the possibility of different pathways in the resubmission.

P5L172: Bias corrections usually assume that the biases at the quantiles do not change over time (stationarity). Thus, the correction itself leads to further errors in the time series, or does it not?

The quantile mapping method used does assume the biases do not change with time. However, we believe this doesn't add any more uncertainty than there would be otherwise from using the raw model data with an inherent bias. Additionally, the simulations only run until 2060, decreasing the any affect a change in bias might have. The effect of the bias correction on the climate data and resultant mass balance was also small.

Model optimisation

P6L189: Why were only 100 random samples generated for parameter optimization? Typically, several hundred or thousand parameter combinations are used.

Three reasons. First, computational limitations. Second, the statistical "rule-of-thumb" when using a Latin hypercube sampling is to have a sample size ten times the number of parameters. Third, we produce multiple parameter combinations that are close to observations, thus the need for further sampling seems unnecessary.

Historical simulations of SMB from climate models (1980-2010)

P8L256: What do you mean exactly with statistically similar?

We conducted T-test comparing the time series of the annual SMB of GFDL and CCSM (p-value=0.0001179)

We will state the statistical test used in the manuscript.

Future SMB of the Juneau Icefield (2031-2060, RCP8.5)

P9L285: Why do the two SMB time series show a very similar trend even though the climate trends and climate sensitivities are so different?

Below are the trends for the future time series of annual mass balance:

GFDL: -0.0712 m w.e. a-1 per year

CCSM: -0.0573 m w.e. a-1 per year

And for the past:

GFDL: -0.0469 m w.e. a-1 per year

CCSM: -0.0609 m w.e. a-1 per year

As you can see, both show a decreasing annual SMB (as expected) but trends do differ between the two simulations. Its clear the GFDL is more sensitive to the climate too with the trend changing the most from the past to the future period.

P9L287: How would the result change to that of Hock et al. (2019) if the change in glacier hypsometry were considered? Wouldn't the results of the two studies then be further apart?

Potentially, as the reviewer suggests, our simulations would be further apart from that of Hock et al. (2019). However, complex feedbacks (glacier detachment, frontal retreat) make this difficult to comment on. In the absence of a coupled ice flow-COSIPY model, we refrain from speculating on this.

P11L322: Why has this physical logic reversed, and snowfall no longer reduces ablation? Is there an explanation for this?

Yes. Although there is high snowfall in these years, the spatial extent is reduced – confined to the highest elevations of glaciers. Without spatially widespread accumulation, the albedo increase of snow cannot counteract melting.

This explanation will be clarified in the revised manuscript.

References:

Collins, M., 2017. Still weighting to break the model democracy. *Geophysical Research Letters*, *44*(7), pp.3328-3329.

Flato, G., Marotzke, J., Abiodun, B., Braconnot, P., Chou, S. C., Collins, W., Cox, P., Driouech,F., Emori, S., Eyring, V., Forest, C., Gleckler, P., Guilyardi, E., Jakob, C., Kattsov, V., Reason, C., and Rummukainen, M.: Evaluation of climate models, Climate Change 2013 – The Physical Science Basis: Working Group I Contribution to the Fifth Assessment Report of the Intergovernmental Panel on Climate Change, Cambridge University Press, 741–866, https://doi.org/10.1017/CBO9781107415324.020, 2013.

Lader, R., Bidlack, A., Walsh, J.E., Bhatt, U.S. and Bieniek, P.A.: Dynamical downscaling for southeast Alaska: Historical climate and future projections, J. App. Meteo. Clim., 59(10), 1607-1623, https://doi.org/10.1175/JAMC-D-20-0076.1, 2020.

Sauter, T., Arndt, A., and Schneider, C.: COSIPY v1.3 – an open-source coupled snowpack and ice surface energy and mass balance model, Geosci. Model Dev., 13, 5645–5662, https://doi.org/10.5194/gmd-13-5645-2020, 2020.

Knutti, R., J. Sedlacek, B. Sanderson, R. Lorenz, E. Fischer, and V. Eyring (2017), A climate model projection weighting scheme accounting for performance and interdependence, *Geophys. Res. Lett.*, **44**, 1909– 1918

Walsh, J.E., Bhatt, U.S., Littell, J.S., Leonawicz, M., Lindgren, M., Kurkowski, T.A., Bieniek, P.A., Thoman, R., Gray, S. and Rupp, T.S.: Downscaling of climate model output for Alaskan stakeholders. Environ. Model. Soft., 110, 38-51, https://doi.org/10.1016/j.envsoft.2018.03.021, 2018

Wendler, G., Gordon, T., & Stuefer, M. (2017). On the Precipitation and Precipitation Change in Alaska. Atmosphere, 8(12), 253. https://doi.org/10.3390/atmos8120253